# Effect of Water Supply on Physiological Response and Phytonutrient Composition of Chili Peppers

**Stella Agyemang Duah** [1,*] , **Clarice Silva e Souza** [1] , **Zsuzsa Nagy** [1] , **Zoltán Pék** [1] , **András Neményi** [1] , **Hussein G. Daood** [2] , **Szergej Vinogradov** [3] **and Lajos Helyes** [1]

1. Institute of Horticulture, Faculty of Agriculture and Environmental Sciences, Hungarian University of Agriculture and Life Sciences, 2100 Gödöllő, Hungary; clarice.souza@phd.uni-szie.hu (C.S.eS.); nagyzsuzsi87@yahoo.com (Z.N.); pek.zoltan@mkk.szie.hu (Z.P.); nemenyi.andras@mkk.szie.hu (A.N.); helyes.lajos@mkk.szie.hu (L.H.)
2. Regional Knowledge Centre, Hungarian University of Agriculture and Life Sciences, 2100 Gödöllő, Hungary; daood.hussein@fh.szie.hu
3. Department of Economics and Methodology, Hungarian University of Agriculture and Life Sciences, 2100 Gödöllő, Hungary; vinogradov.szergej@uni-mate.hu
* Correspondence: agyemang.duah.stella@phd.uni-szie.hu

**Abstract:** Water supply is a primary contributor to the growth and phytonutrient composition in chili peppers. Several physiological stress factors can influence phytonutrients in chili peppers, resulting in their differential synthesis. Maintaining the right and exact amount of water through a drip system can promote an effective fruit set and crop quality. Four pepper cultivars ('Hetényi Parázs'; HET, 'Unikal'; UNIK, 'Unijol'; UNIJ and 'Habanero'; HAB) were investigated under different water supply treatments: RF or rain-fed, DI or deficit irrigation, and OWS or optimum water supply. The two-year experiment was carried out in May 2018 and 2019 under open field conditions. Physiological parameters (relative chlorophyll content, chlorophyll fluorescence, and canopy temperature) were measured during the growth stage and phytonutrients (vitamin C, capsaicinoids and carotenoids) analyzed using high-performance liquid chromatography (HPLC) at harvest in September. The study revealed that, due to higher precipitation and rainfall interruption, increased water supply affected physiological response and phytonutrients in the cultivars. HAB under OWS had a lower response during the growth period when compared to HET, UNIK, and UNIJ. As water supply increased, measured individual carotenoid concentration increased in some cultivars. On the other hand, as water supply decreased, vitamin C and capsaicinoids concentration increased. Even though cultivars responded to the water supply treatments differently, HET exhibited a more uniform and stable composition in all treatments.

**Keywords:** *Capsicum* sp.; physiological response; water supply; phytonutrients

## 1. Introduction

Peppers, either sweet or hot, are known as paprika in Hungary, and they constitute a significant part of traditional and modern cultural cuisine [1]. There are many factors that affect crop productivity, of which the environment or climate and irrigation are among the primary contributors to growth and are expected to influence crop quality. In addition to physiological stress factors, extreme temperatures such as heat, cold, waterlogging stress, and drought can affect the composition of phytonutrients in chili peppers. Chemical features of the soil (moisture and nutrients content) may also stress plants and cause anatomical and physiological disorders. Photosynthetic activities are influenced by factors such as leaf position, stage of development, light intensity, and composition [2,3]. High temperatures and water deficit may cause oxidative stress in leaf water potential and modify the synthesis of carotenoids in pepper plants that grow in stressful environments [4]. The chili pepper crop needs the right and sufficient amount of water for good fruit set and quality [5,6].

Drip irrigation is an effective technological measure in vegetable crop production as it enables the uniform application of water and nutrients, which directly reduces water loss due to evapotranspiration [7–9]. The use of regulated water (deficit irrigation), which is one way of maximizing water use efficiency (WUE), is done by exposing crops to a certain level of water stress either during a particular period (phenological stages) or throughout the whole growing season. However, extreme water stress may directly affect crop quality and phytonutrient concentration [10]. The consideration of deficit irrigation application may be beneficial to water shortage locations without affecting crop quality [11].

Phytochemical concentration in pepper fruits has considerable amounts of vitamins and carotenoid components that are influenced by genotype, stage of maturity, and environmental and postharvest conditions [12,13]. In addition, pepper contains various antioxidants, including capsaicinoids [14,15] and flavonoids, a phenolic compound [16]. Capsaicinoids compounds are known for their therapeutic effectiveness in treating rheumatoid arthritis [17], gastric ulcers [18], anti-obesity, and inflammation [19]. The composition of Vitamin C in pepper promotes collagen production, absorption of inorganic iron, the reduction of plasma cholesterol levels, and strengthening of the immune system [20]. Carotenoid concentration produces the different colors and aroma present in peppers due to the composition of capsanthin, capsorubin, and cryptocapsin [21]. Fat-soluble carotenoids protect the body against cancers, anti-aging, and stimulate the immune system [22].

Studies have been conducted on the irrigation of peppers; however, research on water supply treatments on Hungarian pepper cultivars is limited. The effective use of water through drip irrigation is an excellent water-saving practice that is highly encouraged. As a result of the change in climatic conditions and current inconsistencies in weather forecast predictions, it is necessary to consider the amount of water that pepper plants receive under an open field environment to avoid water losses due to overirrigation.

The aim of the study was to determine the effect of precipitation and irrigation treatments on the phyto-nutritional composition of chili pepper (*Capsicum* sp.) genotypes cultivated under open field conditions using high-performance liquid chromatography (HPLC). In addition, the effect of precipitation and irrigation treatments on the physiological response of the chili cultivars was investigated.

Based on the results of the study, recommendations on suitable water supply for the various pepper cultivars in light of their ability to withstand stress conditions without compromising on their phytochemical composition have been transferred to the breeders, producers, and food industries in Hungary and other countries.

## 2. Materials and Methods

### 2.1. Experimental Conditions

The research was conducted at the Horticulture Institute experimental field, Hungarian University of Agriculture and Life Sciences, Gödöllő, Hungary (latitude 47°61′ N, long. 19°32′ E) with annual average precipitation of around 560 mm. The soil texture was characterized as sandy loam, mostly cambisols with 65% of sand, 8% of clay, 27% of silt fraction, and 1.6% organic matter. The soil had a slight to moderately alkaline pH of 7.9, 16% field capacity, and bulk density of 1.54 g m$^{-3}$ when a depth of about 35 cm of the upper layer of the soil was considered.

In the 2-year experiment conducted from May to September in 2018 and 2019, the same chili pepper cultivars—'Hetényi Parázs' (HET), 'Unikal' (UNIK), 'Unijol' (UNIJ) and 'Habanero' (HAB) were used. Seedlings were obtained from Univer Product Zrt (Kecskemet, Hungary), the leading food industry firm in Hungary.

After 40 days of germination in a nursery, the seedlings were transported for open field cultivation on 17 May 2018 and 13 May 2019, each season. The seedlings were cultivated in twin rows with 0.25 m spacing inside the rows and 0.25 m between plants in a row, with a plant density of 6.66 plants m$^{-2}$ for HET and UNIK. In the case of UNIJ and HAB, the seedlings were planted with a spacing of 0.5 m inside the rows and 0.5 m between plants

in a row, with a plant density of 3.33 plants m$^{-2}$. The spacing between adjacent twin rows of all cultivars was 0.75 m in 2018 and 1.5 m in 2019. The adjusted spacing between twin rows in 2019 was purposely done to manage weed growth easily. The entire experiment was arranged in a randomized complete block design (RCBD) with 4 replicates or blocks per each treatment on a 1-hectare plot of land.

### 2.2. Irrigation System and Management

Irrigation was set up using a drip system for both experimental seasons. A pressure gauge and water meter were installed with control valves in each treatment to manually adjust the water pressure, depth of water supply, and uniformity of water and distribution.

The crop water requirement (ETc) was measured based on the AquaCrop model by Food and Agriculture Organization to determine evapotranspiration (ETo) using the Penman–Monteith method corrected by a crop coefficient (Kc) [23,24]. At each experimental season, weather predictions by the Hungarian Meteorological Services from a nearby station were taken into consideration. The daily minimum and maximum meteorological variables—temperature, relative humidity, and precipitation were calculated. The chili cultivars were given 3 different water supply treatments; rain-fed (RF) except for natural precipitation with no regular irrigation, 50% deficit irrigation (DI), and 100% optimum water supply (OWS) (Table 1, Figure 1).

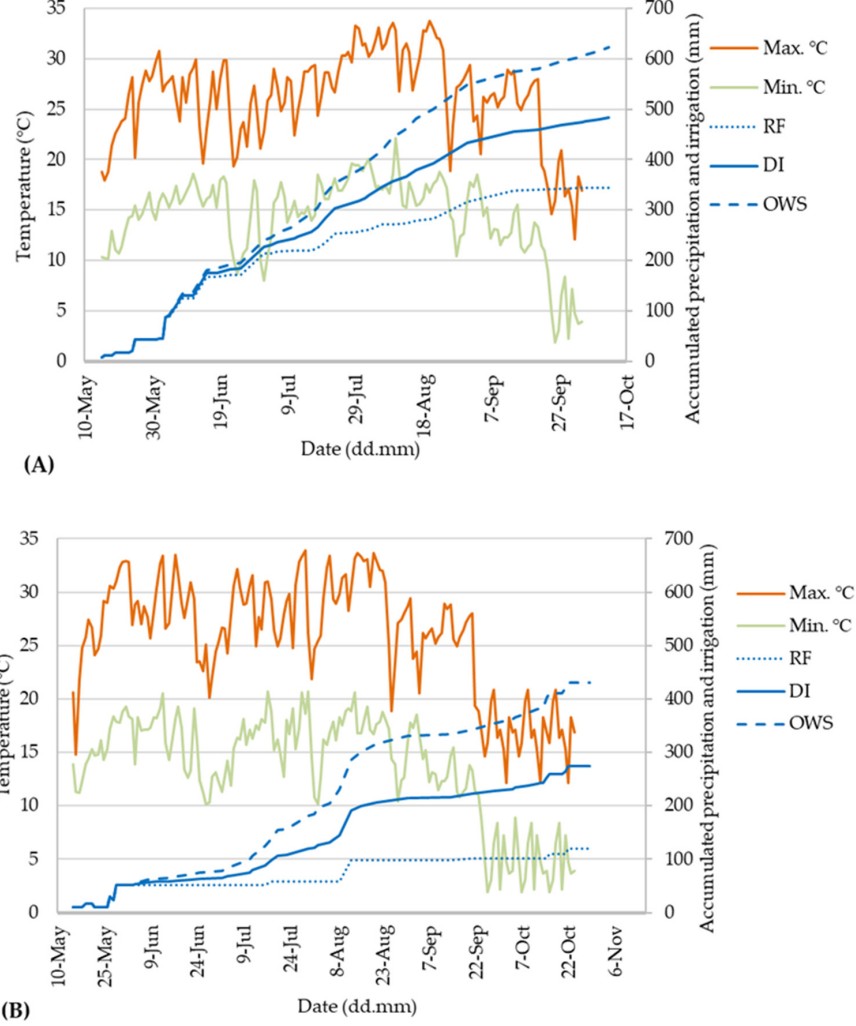

**Figure 1.** Trends of daily maximum (Max.) and minimum (Min.) air temperature (°C), and accumulated precipitation and irrigation (mm) for the growing seasons, dated 5 May 2018–17 October 2018 (**A**) and 5 May 2019–11 November 2019 (**B**).

**Table 1.** Meteorological record and water supply throughout the chili pepper growing seasons.

| Year | Mean Temperature (°C) | Mean Relative Humidity (%) | Precipitation (mm) | Irrigation (mm) | | Total Water Received by Plant Stands (mm) [1] | | |
|---|---|---|---|---|---|---|---|---|
| | | | | DI | OWS | RF | DI | OWS |
| 2018 | 23.8 | 71.0 | 347.8 | 132.6 | 272.2 | 347.8 | 480.4 | 620.0 |
| 2019 | 25.8 | 72.3 | 132.6 | 152.2 | 289.0 | 132.6 | 284.4 | 421.6 |

[1] RF, rain-fed; DI, deficit irrigation; OWS, optimum water supply.

In 2018 between August and early October, the rainfall pattern changed with unexpected heavy rains recorded. The high precipitation in 2018 showed lower mean temperature and relative humidity compared to 2019 (Table 1). During the heavy rainfall period, the crop coefficient (Kc) guidance was considered, and regular irrigation was paused. Regular irrigation of plants was resumed 5 or 6 days after the rains. Generally, irrigation of plants was done 2 times per week depending on precipitation, and once a week, plants received uniform fertilization in the form of granulates proportion of nitrogen ($NO_3$), phosphorus ($P_2O_5$), and potassium ($K_2O$) YaraMila Complex 12-11-18 + 20% sulfur ($SO_3$) (Yara and Co., Veszprem, Hungary).

### 2.3. Measurements and Harvest

During the plant growth periods (2018 and 2019), relative chlorophyll content (expressed as SPAD values), leaf chlorophyll fluorescence (Fv/Fm), and canopy temperature (°C) of newly emerged leaves on plants were measured. Plants were randomly selected in each block per treatment (RF, DI, and OWS).

Fresh ripe peppers, excluding those with any injury to the epidermis, were harvested by hand on 3 September 2018 and 10 September 2019 for chemical analyses.

### 2.4. Physiological Responses

At the time of flowering and harvesting of the peppers, the SPAD index was determined using a chlorophyll meter SPAD-502 (Minolta, Warrington, UK), which measured the greenness of leaves based on the absorbance of 650 nm wavelengths of light, using a reference of 940 nm wavelength infrared light [25]. Only fully expanded new leaves from the apex to the plant base were selected in this experiment. Four plants were randomly selected per block, and in each plant, 4 leaves were measured. In all, 16 leaves per treatment of all cultivars were measured. The chlorophyll meter was calibrated before every measurement.

Chlorophyll fluorescence indicates the physiological health of plants and detects a stress response. A portable PAM 2500 fluorometer (Walz-Mess und Regeltechnik, Germany) was used in this experiment. Measurement was done weekly on sunny days at noon during the entire study period. Four fully developed top leaves of a single plant from each block in all cultivars were affixed with leaf clips for a 35 min dark adaption before fluorescence was measured. Using the Fv/Fm ratio, the maximum quantum efficiency of PSII was quantified and determined by the fast kinetics method in the PamWin 3.0 software [26].

Chlorophyll fluorescence equation:

$$Fv/Fm = (Fm - Fo)/Fm$$

where:

Fo = initial fluorescence
Fm = maximal fluorescence
Fv = variable fluorescence (Fm − Fo).

Canopy temperature reflects the physiological activity of plants, and their growth can be monitored by measurement. A Raytek infrared remote thermometer (Raytek Corporation, Santa Cruz, CA, USA) was used in this experiment. This battery-powered instrument

is capable of measuring 99% of the energy emitted by plants in the field of view of the telemetry unit with an error of $\pm 1\%$, which makes it possible to determine the surface temperature in the plants. In all blocks, 10 plants canopy per treatment of all cultivars were randomly selected in this experiment, and the temperature was recorded. No calibration was required before using the instrument; however, environmental factors, especially clouds, were considered while using the instrument.

### 2.5. Chemical Analyses of Phytonutrients

The vitamin C content was determined according to protocols of Nagy et al. [27] with some modifications. About 3 g of homogenized pepper fruit was crushed in a crucible mortar with quartz sand. 30 mL of 3% metaphosphoric acid solution was gradually added to the mixture and then transferred into a 100 mL Erlenmeyer flask with a stopper.

The mixture was filtered through a filter paper and further purified by passing it through a 0.45 μm cellulose acetate (Whatman, Maidstone, UK) syringe filter before it was injected into an HPLC column. For the quantitative determination of ascorbic acid, sample data were compared to that generated using standard materials (Sigma-Aldrich, Budapest, Hungary).

Total capsaicinoid concentration was determined and calculated as the sum of individual compounds that appeared on the chromatogram following methods of Daood et al. [28]. About 3 g of homogenized pepper fruit were crushed in a crucible mortar with quartz sand. 50 mL of analytical grade methanol was gradually added before the mixture was carefully transferred into a 100 mL Erlenmeyer flask with a stopper. The mixture was subjected to ultrasonication in an ultrasonic bath device for 3 min and then filtered through a filter paper. The filtrate was subjected to 10 times (9:1) dilution process of 9 mL of chromatography grade methanol: 1 mL filtrate for HET and UNIK and purified through a 0.22 μm PTFE (Chromfilter) syringe into vails. UNIJ and HAB were subjected to 20 times (9:1, 1:1) dilution process of 9 mL chromatography grade methanol: 1 mL filtrate and filtered through a 25 μm syringe into a 10 mL glass beaker. The filtrate was further diluted using an Eppendorf pipette of 1 mL methanol and 1 mL filtrate (from syringe filter) into vails. UNIJ and HAB were given 5 times more dilution than the other peppers due to peaks exceeding maximum detection during preliminary analyses. Over this scale, the linearity of the calibration curve will be lost. Therefore, extra dilution was needed to have peaks below the maximum detection scale of the FL detector. The capsaicinoid peaks that were identified on the chromatogram were nordihydrocapsaicin (NDC), capsaicin (CAP), dihydrocapsaicin (DC), homocapsaicin (HCAP), dihydrocapsaicin isomer (iDC), and sum of homodihydrocapsaicin (HDCs).

Total carotenoids were determined according to the methods and protocols of Daood et al. [29] with modifications. About 2.5 g of homogenized pepper fruit from HET, UNIK, UNIJ, and 3.5 g from HAB (due to their orange-red color attribute) was crushed in a crucible mortar with quartz sand. 20 mL of methanol was added for 1–2 min, and the upper layer poured into an Erlenmeyer flask. 10 mL of methanol (analytical grade) was then added to 50 mL of dichloroethane in a 100 mL graduated cylinder and shaken gently.

The mixture was poured into the remaining homogenized pepper in the crucible mortar and then transferred into the Erlenmeyer flask and shaken vigorously. A few drops of distilled water were added and shaken gently. The mixture was separated with a burette into a flat bottom flask using a filter paper containing sodium sulfate anhydrous in a separating funnel. 5 mL of dichloroethane was added to the filtrate through a filter paper for further extraction and evaporated with a rotary evaporation chamber for 10 min at 70 °C and 40 °C vacuum, respectively.

The flask was offloaded from the chamber tube after all filtrate evaporated. 5 mL of pigment eluents and 5 mL of methanol (analytical grade) were respectively dropped into the flask and shaken evenly for HET, UNIK, and UNIJ. In the case of HAB, 2.5 mL of pigment eluents and 2.5 mL of methanol (analytical grade) were respectively dropped

into the flask and shaken evenly. An ultrasonic shaker was used where necessary to ensure that no residue was left in the flask. Then the solution was filtered through a 0.22 μm PTFE membrane syringe into the veils and injected into the HPLC column. The individual carotenoid peaks that were identified and analyzed on the chromatogram were free capsanthin (free caps), free zeaxanthin (free zeax), capsanthin mono-ester (caps ME), zeaxanthin mono-ester (zeax ME), Beta-carotene (β-carotene), capsanthin di-ester (caps DE), and zeaxanthin di-ester (zeax DE).

In all phytonutrient analyses, an HPLC (Hitachi Chromaster, Tokyo, Japan) instrument consisting of a Model 5110 Pump, a Model 5210 Auto Sampler, a Model 5430 Diode Array detector, and a Model 5440 FL detector, was used for the determination of all compounds. All chemicals, including analytical and HPLC grade solvents, were obtained from VWR (Budapest, Hungary, and Darmstadt, Germany).

*2.6. Statistical Analysis*

Data were expressed as the mean ± standard deviation (SD) among physiological responses, pepper cultivars, water supply treatments, and phytonutrients. The Kolmogorov–Smirnov test was used to decide if samples come from populations with a normal distribution. Levene's test was used to test the variance's homoscedasticity, where the null hypothesis was that the variances within each of the examined groups were the same.

One-way analysis of variance (ANOVA) was used to examine the effect of water supply (RF, DI, and OWS) on physiological responses (SPAD, chlorophyll fluorescence, and canopy temperature), vitamin C, capsaicinoids (NDC, CAP, DC, HCAP, iDC, HDC-1, and HDC-2) and carotenoids (free caps, free zeax, caps ME, zeax ME, β-carotene, caps DE, and zeax DE). ANOVA was also used to examine significant differences among cultivars (HET, UNIK, UNIJ, and HAB). In the case of a significant result of the ANOVA, the groups with significant differences were determined by Tukey HSD (Honestly Significant Difference) post-hoc test.

All statistical analyses were carried out with IBM SPSS Software package version 25.0 for Windows, at the significance level $\alpha = 0.05$.

**3. Results**

*3.1. Effect of Water Supply on Physiological Responses of Cultivars during the Growth Period*

During the 2018 and 2019 growth period, the chili pepper cultivars (HET, UNIK, UNIJ, and HAB) were given different water supply treatments (RF, DI, and OWS), and physiological parameters were measured. In the first growing season (2018), plant stands received more water due to rain (varied between 348–620 mm) when compared to the 2019 growing season (varied between 133–422 mm), which was mildly dry all through (Figure 1).

In 2018 (Figure 2A), water supply had no significant influence on HET and UNIK even though a slight decrease in SPAD values was recorded in DI and OWS, compared to RF. In both UNIJ and HAB, lower SPAD values were recorded in OWS when compared to RF. HAB had the lowest SPAD values significantly among all cultivars (F = 35.357, $p < 0.001$). UNIK recorded the highest relative chlorophyll content in all cultivars but was not significantly different from HET.

Similarly, all cultivars in 2019 (Figure 2B) had significant differences among them. There was no significant effect on water supply in HET cultivar (F = 0.547, $p = 0.582$). UNIK recorded significantly ($p < 0.001$) the highest SPAD values. Under OWS conditions, UNIK recorded significantly lower SPAD values compared to that of RF. As the water supply increased, SPAD content decreased in UNIJ. In addition, in HAB, a decrease in SPAD values as the water supply increased was detected. Peppers irrigated (OWS) recorded the lowest SPAD values and in the non-irrigated ones (RF) the highest; however, DI was not significantly higher when compared to OWS (F = 17.081, $p < 0.001$).

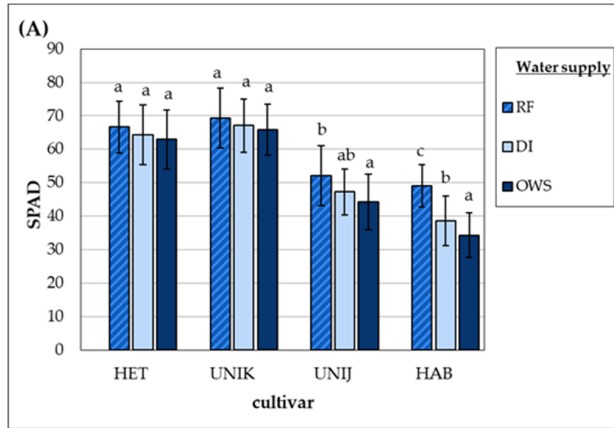
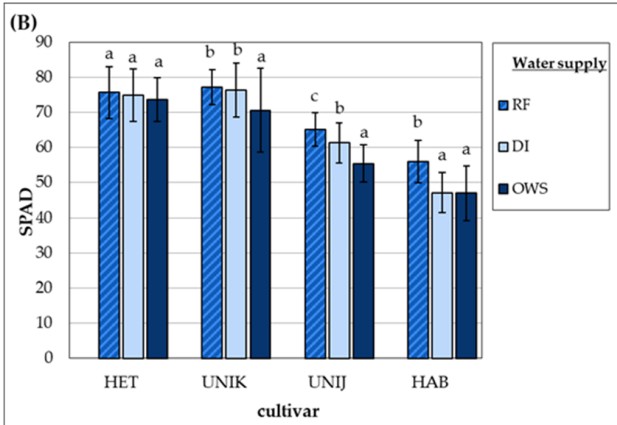

**Figure 2.** Effect of water supply treatments on relative chlorophyll content (expressed as SPAD values) of cultivars in 2018 (**A**) and 2019 (**B**) growing seasons. The data represents the average values of both years ± SD. The values showing different letters are significantly different at $p \leq 0.05$ using the post-hoc HSD test. RF, rain-fed; DI, deficit irrigation; OWS, optimum water supply; HET, Hetényi Parázs; UNIK, Unikal; UNIJ, Unijol; HAB, Habanero.

Values of Fv/Fm in 2018 were in general lower in HAB compared to the other cultivars (Figure 3A). Furthermore, during this year, values were lower in RF compared to those of the other treatments, even if differences were not significant (Figure 3A).

In the second growing season (Figure 3B), HET had significantly ($p = 0.021$) lower Fv/Fm values in DI and OWS when compared to RF (F = 10.101, $p < 0.001$). Among the other cultivars (UNIK, UNIJ, and HAB), Fv/Fm values of water supply treatments were not significantly different from each other. Nonetheless, it was detected that as water supply in HAB, values tendentially increased (F = 2.537, $p = 0.085$).

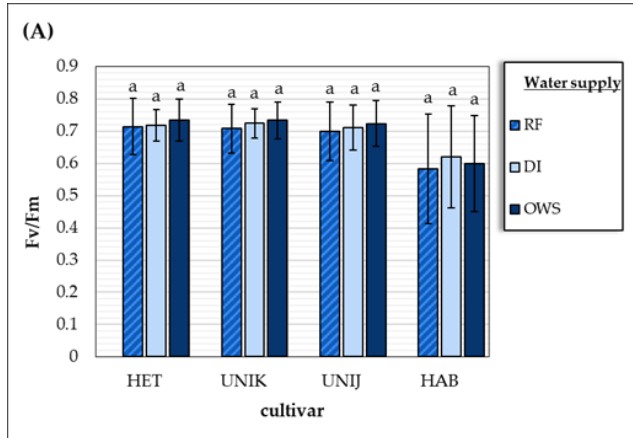
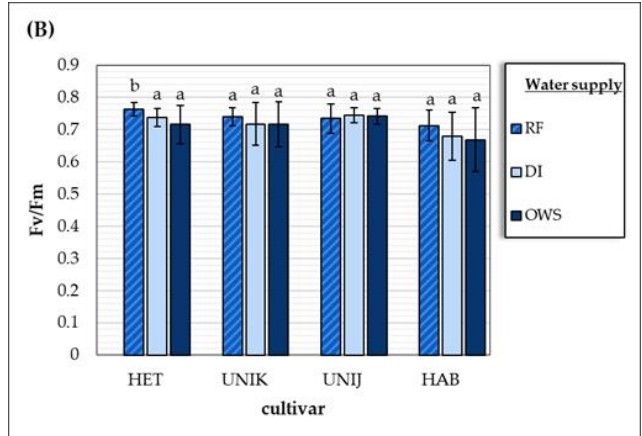

**Figure 3.** Effect of water supply treatments on chlorophyll fluorescence (Fv/Fm) measured in 2018 (**A**) and 2019 (**B**) growing seasons. The data represents the average values of both years ± SD. The values showing different letters are significantly different at $p \leq 0.05$ using post-hoc HSD test. RF, rain-fed; DI, deficit irrigation; OWS, optimum water supply; HET, Hetényi Parázs; UNIK, Unikal; UNIJ, Unijol; HAB, Habanero.

Even though lower canopy temperature was detected under OWS, water supply treatments had no effect on all cultivars during the 2018 growing season (Figure 4A). Nevertheless, a gradual decrease in canopy temperature was recorded in cultivars as the water supply increased.

On the effect of water supply treatments, in the 2019 season (Figure 4B), HET had a significantly ($p = 0.001$) lower response to canopy temperature under DI and OWS conditions when compared to RF. Water supply had no influence on UNIK, UNIJ, and HAB. Notwith-

standing, as water supply increased, canopy temperature increased in UNIJ, but on the contrary, that of HAB decreased even though there were no significant differences in them.

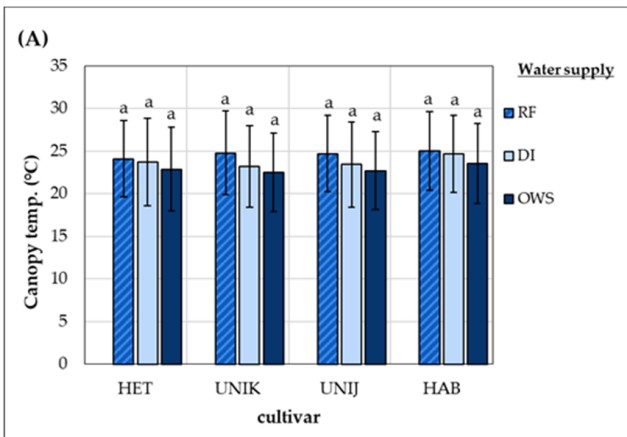 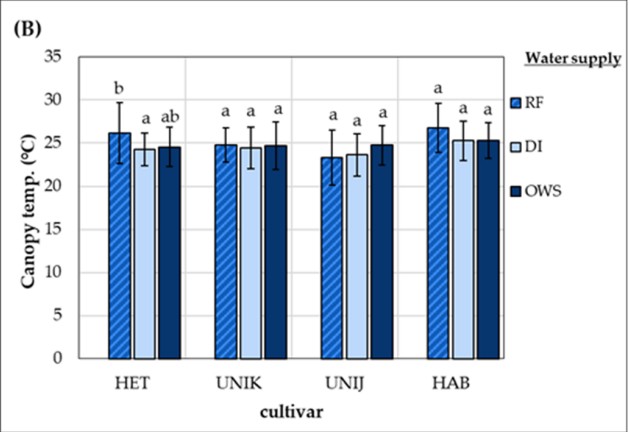

**Figure 4.** Effect of water supply treatments on canopy temperature (°C) of cultivars in 2018 (**A**) and 2019 (**B**) growing seasons. The data represents the average values of both years ± SD. The values showing different letters are significantly different at $p \leq 0.05$ using post-hoc HSD test. RF, rain-fed; DI, deficit irrigation; OWS, optimum water supply; HET, Hetényi Parázs; UNIK, Unikal; UNIJ, Unijol; HAB, Habanero.

### 3.2. Effect of Water Supply on Phytonutrients in Chili Cultivars

The vitamin C content varied greatly depending on the cultivar, the degree of water supply, and the crop year. During the 2018 season (Figure 5), vitamin C content was lower in HET peppers grown under OWS conditions when compared to RF (F = 5.405, $p$ = 0.029). In UNIK, a higher content of vitamin C was recorded in DI when compared to RF and OWS. In UNIJ, OWS was favorable for increased vitamin C content when compared to DI, and in particularly compared to RF ($p < 0.001$). At last, water supply had no influence on vitamin C in HAB even though a slight increase in content was found in DI.

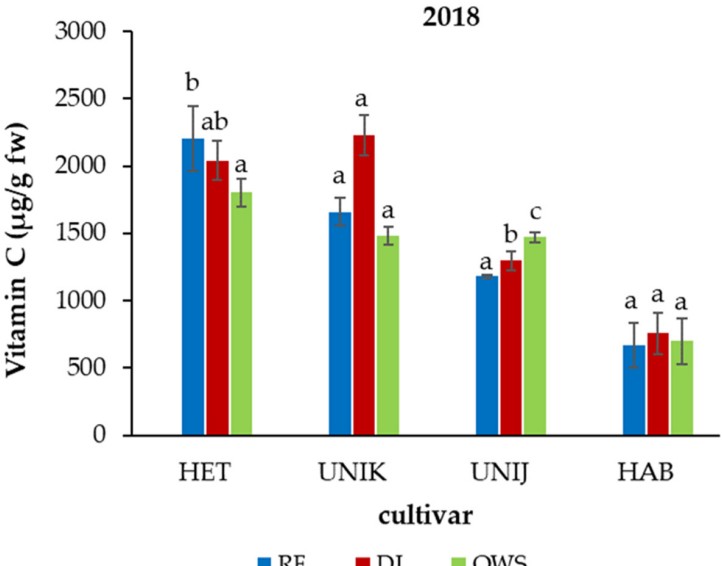

**Figure 5.** Effect of water supply on Vitamin C content in the chili cultivars for 2018 season; expressed in μg/g fresh base weight ± SD (n = 4). The same letter shows no significant difference in vitamin C among water supply treatments of cultivars according to Tukey's HSD post-hoc test. RF, rain-fed; DI, deficit irrigation; OWS, optimum water supply; HET, Hetényi Parázs; UNIK, Unikal; UNIJ, Unijol; HAB, Habanero.

Among the cultivars, HET had the highest amounts of vitamin C and HAB the lowest.

Generally, in all cultivars, as water supply increased, vitamin C content decreased in the 2019 season (Figure 6). HET had a lower vitamin C content under OWS condition when compared to RF (F = 4.804, *p* = 0.038). In UNIK, vitamin C content was found to be higher in RF but was not significantly different from those of DI and OWS. In both UNIJ and HAB, higher amounts of vitamin C content were recorded in RF when compared to DI and OWS (F = 9.832, *p* = 0.005 and F = 18.720, *p* = 0.001), respectively.

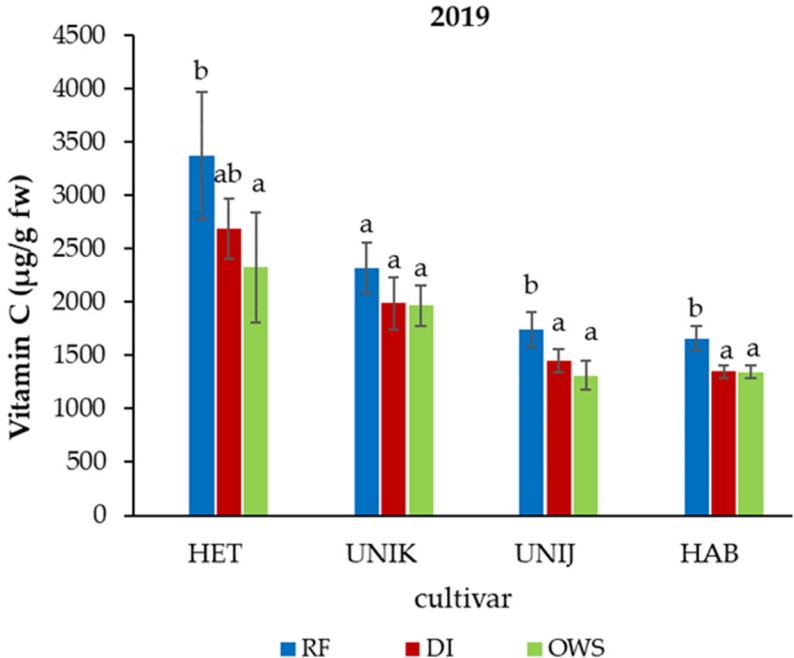

**Figure 6.** Effect of water supply on Vitamin C content in the chili cultivars for 2019 growing seasons; expressed in μg/g fresh base weight ± SD (n = 4). The same letter shows no significant difference in vitamin C among water supply treatments of cultivars according to Tukey's HSD post-hoc test. RF, rain-fed; DI, deficit irrigation; OWS, optimum water supply; HET, Hetényi Parázs; UNIK, Unikal; UNIJ, Unijol; HAB, Habanero.

Among the cultivars, HET recorded the highest vitamin C contents, and UNIJ and HAB the lowest.

The major capsaicinoids compounds (NDC, CAP, and DC) were present in all cultivars in 2018 (Table 2). The concentration of NDC was found to be higher in UNIJ and lowest in UNIK. Water supply had no influence on CAP concentration in HET and HAB even though under DI conditions, these cultivars had higher concentrations and lower in RF. A significantly (*p* = 0.015) higher amounts of CAP were found in UNIK under DI and OWS conditions when compared to RF; furthermore, between DI and OWS, no significant differences were detected.

In the homolog compounds (HCAP, iDC, and HDCs), HCAP was only present in HET and absent in the other cultivars. Furthermore, water supply had no significant influence on HCAP amounts in HET cultivar. Water supply treatments did not affect iDC and HDCs concentration in all cultivars.

Generally, among cultivars, a higher concentration of capsaicinoids was present in HAB and UNIJ and lower in UNIK.

**Table 2.** Effect of water supply on capsaicinoids concentration in the pepper cultivars for the 2018 growing season. The means are expressed in µg/g fresh base weight ± SD (n = 4).

| Cultivar | Water Supply | NDC (µg/g) | CAP (µg/g) | DC (µg/g) | HCAP (µg/g) | iDC (µg/g) | HDCs (µg/g) |
|---|---|---|---|---|---|---|---|
| HET | RF | 9.6 ± 0.2 a | 58.2 ± 14.4 a | 33.6 ± 3.5 a | 1.17 ± 0.1 a | 4.1 ± 0.5 a | 5.32 ± 0.7 a |
| | DI | 9.3 ± 1.8 a | 69.7 ± 11.6 a | 43.7 ± 2.3 a | 1.24 ± 0.1 a | 3.5 ± 0.6 a | 4.17 ± 0.7 a |
| | OWS | 8.9 ± 1.9 a | 62.1 ± 12.2 a | 41.1 ± 4.4 a | 1.2 ± 0.2 a | 3.7 ± 0.7 a | 4.11 ± 0.4 a |
| | F-value | 0.119 | 0.749 | 0.286 | 0.061 | 0.516 | 0.720 |
| | *p* value | 0.889 | 0.500 | 0.758 | 0.941 | 0.614 | 0.513 |
| UNIK | RF | 2.8 ± 0.1 a | 17.3 ± 1.8 a | 13.5 ± 1.8 a | ND | 1.05 ± 0.40 a | 1.0 ± 0.1 a |
| | DI | 4.0 ± 0.3 a | 29.2 ± 2.9 b | 19.8 ± 3.2 a | ND | 1.6 ± 0.35 a | 1.6 ± 0.2 a |
| | OWS | 3.7 ± 0.4 a | 30.3 ± 4.9 b | 18.5 ± 1.2 a | ND | 0.9 ± 0.35 a | 1.7 ± 0.2 a |
| | F-value | 1.360 | 6.908 | 2.640 | | 3.900 | 2.053 |
| | *p* value | 0.305 | 0.015 | 0.125 | | 0.060 | 0.184 |
| UNIJ | RF | 116.4 ± 11.6 a | 1282.7 ± 137.3 a | 796.2 ± 36.9 a | ND | 41.1 ± 3.4 a | 35.1 ± 2.2 a |
| | DI | 117.2 ± 9.3 a | 1239.9 ± 142.3 a | 766.5 ± 61.2 a | ND | 44.6 ± 4.1 a | 36.2 ± 4.8 a |
| | OWS | 112.4 ± 14.5 a | 1213.6 ± 153.4 a | 744.6 ± 71.4 a | ND | 31.0 ± 2.61 a | 33.2 ± 4.2 a |
| | F-value | 0.341 | 0.157 | 0.790 | | 1.033 | 0.103 |
| | *p* value | 0.720 | 0.857 | 0.483 | | 0.394 | 0.903 |
| HAB | RF | 60.3 ± 7.5 a | 1822.1 ± 121.3 a | 996.4 ± 108.6 a | ND | 45.0 ± 5.4 a | 22.7 ± 2.2 a |
| | DI | 79.8 ± 10.3 a | 2191.8 ± 247 a | 1080.6 ± 119.2 a | ND | 43.8 ± 4.40 a | 24.4 ± 5.3 a |
| | OWS | 75.6 ± 13.2 a | 2130.6 ± 216.3 a | 1029.8 ± 189.1 a | ND | 64.8 ± 7.8 a | 25.2 ± 7.5 a |
| | F-value | 0.407 | 0.696 | 0.301 | | 0.694 | 0.221 |
| | *p* value | 0.677 | 0.524 | 0.747 | | 0.524 | 0.806 |

The same letter shows no significant difference in capsaicinoid compounds among water supply treatments of cultivars according to Tukey's HSD post-hoc test; ND: Not detected; NDC: Nordihydrocapsaicin; CAP: Capsaicin; DC: Dihydrocapsaicin; HCAP: Homocapsaicin; iDC: Dihydrocapsaicin isomer; HDCs: Sum of homodihydrocapsaicin; HET, Hetényi Parázs; UNIK, Unikal; UNIJ, Unijol; HAB, Habanero.

During the 2019 season, all major compounds were present in higher concentrations in all cultivars and in minimal amounts in the homologs (Table 3). The concentration of NDC in HET reduced significantly ($p = 0.006$) as water supply treatments increased; however, between DI and OWS, no significant difference was detected. Water supply had no influence on NDC concentration in UNIK, even though higher amounts were found in RF. In UNIJ, higher concentrations of NDC were found in RF and significantly ($p < 0.001$) lower in DI and OWS. A similar trend was observed in HAB; as water supply treatments increased, NDC concentration significantly ($p < 0.001$) decreased; nevertheless, concentration in DI was not different from that of OWS. CAP concentration in HET and UNIK were lower under DI and OWS when compared to RF; however, there were no significant differences among them. A significantly ($p < 0.001$) lower concentration of CAP was found in UNIJ cultivar under OWS when compared to RF and DI. Higher amounts of CAP in HAB were detected in RF and DI and significantly ($p = 0.032$) lower in OWS. Under RF conditions, DC concentrations in HET were observed to be significantly ($p = 0.019$) higher when compared to DI and OWS. A similar observation of DC concentration was found in UNIJ ($p < 0.001$). Water supply had no influence on DC concentration in UNIK, although higher amounts were found in RF. In HAB, as the water supply increased, DC concentration significantly ($p < 0.001$) and progressively decreased.

In the homologs (HCAP, iDC, and HDCs), it was detected that water supply had no influence on HCAP concentration in HET, UNIK, and HAB cultivars and were absent in UNIJ. As the water supply increased, iDC amounts significantly decreased in all cultivars even though concentrations in OWS were not always different from those of DI. A similar trend was observed in HDCs in HET, UNIJ, and HAB; a significantly lower concentration was detected as water supply increased. However, in UNIK, concentration did not change irrespective of water supply treatments.

Between cultivars, HAB had higher capsaicinoids concentration and lower amounts were found in UNIK.

**Table 3.** Effect of water supply on capsaicinoids concentration in the pepper cultivars for the 2019 growing season. The means are expressed in µg/g fresh base weight ± SD (n = 4).

| Cultivar | Water Supply | NDC (µg/g) | CAP (µg/g) | DC (µg/g) | HCAP (µg/g) | iDC (µg/g) | HDCs (µg/g) |
|---|---|---|---|---|---|---|---|
| HET | RF | 42.9 ± 10.5 b | 405.1 ± 67.0 a | 250.5 ± 46.6 b | 2.1 ± 0.3 a | 7.2 ± 1.0 b | 24.4 ± 4.1 b |
| | DI | 27.1 ± 6.4 a | 255.6 ± 29.4 a | 151.0 ± 25.8 a | 1.5 ± 0.1 a | 4.0 ± 1.0 a | 14.1 ± 2.9 a |
| | OWS | 20.3 ± 4.2 a | 238.7 ± 27.7 a | 124.3 ± 20.1 a | 1.5 ± 0.1 a | 2.4 ± 0.40 a | 11.1 ± 1.8 a |
| | F-value | 9.456 | 3.595 | 6.298 | 0.600 | 29.328 | 9.318 |
| | *p* value | 0.006 | 0.071 | 0.019 | 0.569 | ≤0.001 | 0.006 |
| UNIK | RF | 18.2 ± 2.1 a | 232.4 ± 31.1 a | 99.4 ± 18.3 a | 1.8 ± 0.1 a | 2.8 ± 0.3 b | 13.3 ± 1.3 a |
| | DI | 17.8 ± 2.9 a | 131.6 ± 17.3 a | 80.1 ± 9.3 a | 1.5 ± 0.1 a | 2.2 ± 0.3 b | 12.7 ± 2.7 a |
| | OWS | 13.8 ± 2.3 a | 98.2 ± 16.5 a | 58.6 ± 6.8 a | 1.2 ± 0.1 a | 1.7 ± 0.1 a | 7.3 ± 1.1 a |
| | F-value | 1.564 | 3.709 | 1.075 | 3.268 | 8.587 | 2.060 |
| | *p* value | 0.261 | 0.067 | 0.381 | 0.086 | 0.008 | 0.183 |
| UNIJ | RF | 151.4 ± 21.7 b | 1662.5 ± 235.4 b | 1108.6 ± 156.8 b | ND | 39.4 ± 11.2 b | 53.4 ± 5.9 b |
| | DI | 117.8 ± 7.7 a | 1796.7 ± 242.7 b | 1050.5 ± 83.1 a | ND | 23.6 ± 7.8 a | 38.5 ± 1.6 b |
| | OWS | 87.0 ± 18.9 a | 1513.5 ± 139.4 a | 743.7 ± 47.6 a | ND | 16.6 ± 5.2 a | 27.8 ± 1.0 a |
| | F-value | 22.795 | 65.553 | 76.909 | | 7.636 | 9.032 |
| | *p* value | ≤0.001 | ≤0.001 | ≤0.001 | | 0.012 | 0.007 |
| HAB | RF | 108.5 ± 19.6 b | 2744.3 ± 316.9 b | 1323.8 ± 155.8 c | 21.7 ± 1.0 a | 56.0 ± 11.3 b | 51 ± 4.5 b |
| | DI | 63.9 ± 8.3 a | 2969.7 ± 162.0 b | 1034.2 ± 99.4 b | 14.0 ± 1.6 a | 33.5 ± 8.2 a | 28.8 ± 6.6 a |
| | OWS | 42.9 ± 3.4 a | 2392.2 ± 262.6 a | 742.8 ± 75.9 a | 9.8 ± 0.1 a | 22.3 ± 1.3 a | 22.3 ± 1.0 a |
| | F-value | 29.081 | 5.188 | 25.357 | 0.500 | 9.831 | 27.071 |
| | *p* value | ≤0.001 | 0.032 | ≤0.001 | 0.622 | 0.005 | ≤0.001 |

The same letter shows no significant difference in capsaicinoid compounds between water supply treatments of cultivars according to Tukey's HSD post-hoc test; ND: Not detected; NDC: Nordihydrocapsaicin; CAP: Capsaicin; DC: Dihydrocapsaicin; HCAP: Homocapsaicin; iDC: Dihydrocapsaicin isomer; HDCs: Sum of homodihydrocapsaicin; HET, Hetényi Parázs; UNIK, Unikal; UNIJ, Unijol; HAB, Habanero.

The water supply effect on carotenoid compounds during the 2018 season was identified on the chromatogram and assessed (Table 4). HET under RF conditions had the highest free caps concentration compared to OWS but was not significantly different from that of DI ($p = 0.009$). In UNIK, water supply treatments did not influence free caps concentration. Free caps amount significantly ($p = 0.007$) decreased in UNIJ under OWS compared to RF conditions even though concentration did not differ from that of DI. HAB had free caps concentration with significantly ($p < 0.001$) higher amounts found in RF when compared to DI and OWS. Water supply treatments did not affect caps ME amount in UNIK and HAB. However, in HET, caps ME concentration decreased significantly ($p = 0.004$) under OWS condition RF and DI. In UNIJ; as water supply increased, caps ME concentration decreased significantly ($p < 0.001$) even though between DI and OWS, no significant difference was found. The application of water supply treatments did not influence caps DE concentration in HET, UNIK, and HAB, while in UNIJ, as water supply increased, caps DE amount significantly ($p = 0.002$) decreased in DI and OWS.

Water supply did not change the concentration of free zeax in HET, UNIK, and UNIJ; however, in HAB, the concentration decreased significantly ($p = 0.009$) below detection limit as water supply increased. Considering the monoesters of zeaxanthin (zeax ME and zeax DE), water supply did not affect concentration in UNIK and HAB. However, it was detected in UNIJ that as water supply increased, zeax ME amounts significantly ($p = 0.011$) decreased under OWS even though between DI and OWS, concentration did not change. Water supply treatment had no influence on zeax DE concentration in UNIJ.

On the effect of water supply on β-carotene in the chili cultivar, concentration did not change in HET, UNIK, and HAB. Nonetheless, in UNIJ, higher amounts of β-carotene were detected in RF and significantly ($p = 0.001$) lower in DI and OWS

**Table 4.** Effect of water supply on carotenoid concentration in the 2018 season. The means are expressed in µg/g fresh base weight ± SD (n = 4).

| Cultivar | Water Supply | Free Caps. (µg/g) | Free Zeax. (µg/g) | Caps ME (µg/g) | Zeax ME (µg/g) | β-Carotene (µg/g) | Caps DE (µg/g) | Zeax DE (µg/g) |
|---|---|---|---|---|---|---|---|---|
| HET | RF | 36.9 ± 5.4 b | 4.1 ± 1.5 a | 54.5 ± 6.5 b | 7.6 ± 0.7 a | 40.8 ± 5.0 a | 215.7 ± 3.3 a | 59.6 ± 1.7 b |
| | DI | 35.4 ± 2.9 b | 2.3 ± 0.3 a | 53.7 ± 5.5 b | 8.0 ± 0.7 a | 36.3 ± 0.6 a | 225.3 ± 22.5 a | 45.2 ± 0.9 b |
| | OWS | 26.5 ± 2.7 a | 2.5 ± 0.9 a | 37.6 ± 5.2 a | 8.6 ± 2.7 a | 28.7 ± 3.2 a | 148.1 ± 16.4 a | 34.7 ± 1.2 a |
| | F value | 8.279 | 3.680 | 11.058 | 0.312 | 0.958 | 2.654 | 84.118 |
| | *p* value | 0.009 | 0.068 | 0.004 | 0.740 | 0.419 | 0.124 | ≤0.001 |
| UNIK | RF | 42.8 ± 7.6 a | 6.4 ± 0.8 a | 45.2 ± 6.8 a | 11.3 ± 2.1 a | 38.4 ± 2.7 a | 128.3 ± 12.4 a | 33.6 ± 2.0 a |
| | DI | 56.6 ± 8.5 a | 5.9 ± 0.7 a | 49.4 ± 5.2 a | 15.0 ± 1.3 a | 28.9 ± 4.7 a | 126.0 ± 16.8 a | 19.7 ± 3.3 a |
| | OWS | 32.3 ± 3.1 a | 3.8 ± 0.4 a | 38.7 ± 3.5 a | 8.2 ± 1.2 a | 34.7 ± 4.5 a | 120.4 ± 23.2 a | 15.9 ± 0.6 a |
| | F value | 1.359 | 1.503 | 0.699 | 0.770 | 0.152 | 0.176 | 0.739 |
| | *p* value | 0.305 | 0.273 | 0.522 | 0.491 | 0.861 | 0.842 | 0.504 |
| UNIJ | RF | 51.2 ± 5.6 b | 8.8 ± 1.8 a | 54.2 ± 7.1 b | 23.0 ± 3.5 b | 64.1 ± 2.7 b | 177.8 ± 16.9 b | 58.5 ± 9.4 a |
| | DI | 24.7 ± 3.3 a | 5.9 ± 1.7 a | 31.9 ± 3.7 a | 15.7 ± 2.5 a | 40.1 ± 1.6 a | 114.3 ± 12.2 a | 42.8 ± 4.6 a |
| | OWS | 23.0 ± 3.5 a | 6.7 ± 0.8 a | 29.4 ± 4.3 a | 17.1 ± 2.2 a | 37.1 ± 1.9 a | 114.9 ± 19.9 a | 39.9 ± 5.6 a |
| | F value | 9.079 | 0.836 | 22.890 | 7.809 | 18.957 | 13.558 | 2.063 |
| | *p* value | 0.007 | 0.464 | ≤0.001 | 0.011 | 0.001 | 0.002 | 0.183 |
| HAB | RF | 1.9 ± 0.2 b | 0.4 ± 0.0 b | 1.2 ± 0.1 a | ND | 5.3 ± 2.4 a | 3.6 ± 2.1 a | 2.2 ± 0.3 a |
| | DI | 0.6 ± 0.1 a | 0.1 ± 0.0 a | 0.7 ± 0.2 a | ND | 6.5 ± 2.9 a | 4.9 ± 2.3 a | 1.1 ± 0.1 a |
| | OWS | 0.6 ± 0.1 a | ND | 0.5 ± 0.1 a | ND | 6.6 ± 1.2 a | 5.7 ± 1.6 a | 1.3 ± 0.1 a |
| | F value | 77.448 | 8.203 | 2.433 | | 2.625 | 1.066 | 1.177 |
| | *p* value | ≤0.001 | 0.009 | 0.143 | | 0.126 | 0.384 | 0.351 |

The same letter shows no significant difference in carotenoid concentration between water supply treatments of cultivars according to Tukey's HSD post-hoc test; ND: Not detected; free caps: Free capsanthin; free zeax: Free zeaxanthin; caps ME: Capsanthin mono-ester; zeax ME: Zeaxanthin mono-ester; β-carotene: Beta-carotene; caps DE: Capsanthin di-ester; zeax DE: Zeaxanthin di-ester; HET, Hetényi Parázs; UNIK, Unikal; UNIJ, Unijol; HAB, Habanero.

The sum of all individual carotenoid peaks identified on the chromatogram showed that water supply had an influence on some cultivars (Figure 7). In HET and UNIK, a slight decrease of concentration was observed as water supply increased; however, no significant change was seen. Meanwhile, under RF conditions in UNIJ and HAB, higher concentration of carotenoids was detected when compared to DI and OWS (F = 19.984, *p* < 0.001 and F = 5.670, *p* = 0.025, respectively). Between cultivars, UNIJ had the highest carotenoid concentration, and HAB recorded the lowest amount.

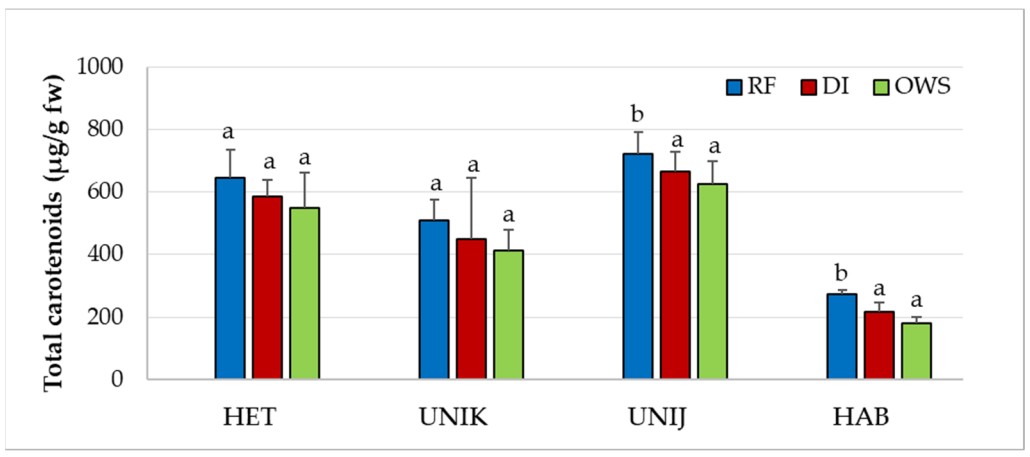

**Figure 7.** Effect of water supply on total carotenoid concentration in the 2018 season. The means are expressed in µg/g fresh base weight ± SD (n = 4). RF, rain-fed; DI, deficit irrigation; OWS, optimum water supply; HET, Hetényi Parázs; UNIK, Unikal; UNIJ, Unijol; HAB, Habanero.

The effect of water supply on individual carotenoid concentration in the chili cultivars for the 2019 season was assessed (Table 5). An increase in water supply significantly ($p$ = 0.007) lowered free caps concentration in HET even though between DI and OWS, no change in concentration was recorded. It was observed in UNIK and UNIJ cultivars that water supply had no influence on free caps concentration. HAB, on the other hand, had a significant ($p < 0.001$) decrease in free caps concentration as water supply increased, and under OWS condition, no free caps was detected. Caps ME amount in HET was found to be significantly ($p$ = 0.021) lower under DI when compared to RF. In addition, in HAB, caps ME concentration were significantly ($p$ = 0.041) lower under DI when compared to RF and OWS. As water supply increased, caps ME concentration in UNIK significantly ($p$ = 0.007) increased in DI and OWS when compared to RF. In UNIJ, water supply treatments did not influence caps ME concentration. Considering caps DE concentration in HET, under RF conditions, a significantly ($p$ = 0.029) higher amount was recorded when compared to DI. A similar trend was recorded in UNIJ; caps DE concentration was significantly ($p < 0.001$) higher in RF when compared to DI. In UNIK, water supply did not influence caps DE amount. Caps DE amount in HAB increased significantly ($p$ = 0.006) under DI and OWS conditions when compared to RF.

**Table 5.** Effect of water supply on carotenoid concentration in the 2019 season. The means are expressed in µg/g fresh base weight ± SD (n = 4).

| Cultivar | Water Supply | Free Caps. (µg/g) | Free Zeax. (µg/g) | Caps ME (µg/g) | Zeax ME (µg/g) | β-Carotene (µg/g) | Caps DE (µg/g) | Zeax DE (µg/g) |
|---|---|---|---|---|---|---|---|---|
| HET | RF | 1.9 ± 0.4 b | 15.1 ± 3.5 b | 57.5 ± 6.1 b | 59.6 ± 1.7 a | 41.9 ± 2.8 a | 256.3 ± 42.0 b | 51.2 ± 5.2 a |
|  | DI | 1.0 ± 0.3 a | 8.8 ± 2.0 a | 33.9 ± 3.0 a | 45.2 ± 0.9 a | 31.7 ± 0.8 a | 152.1 ± 31.6 a | 34.2 ± 2.8 a |
|  | OWS | 1.2 ± 0.4 a | 9.4 ± 3.9 a | 37.7 ± 5.8 ab | 34.7 ± 1.2 a | 31.7 ± 4.0 a | 171.7 ± 31.6 ab | 38.1 ± 8.0 a |
|  | F value | 9.196 | 4.559 | 6.091 | 3.731 | 2.926 | 5.416 | 2.314 |
|  | *p* value | 0.007 | 0.043 | 0.021 | 0.066 | 0.105 | 0.029 | 0.155 |
| UNIK | RF | 1.1 ± 0.5 a | 10.6 ± 2.4 a | 27.0 ± 3.7 a | 33.6 ± 2.0 a | 40.2 ± 4.5 a | 132.8 ± 21.7 a | 13.3 ± 2.1 a |
|  | DI | 1.0 ± 0.2 a | 8.7 ± 0.9 a | 52.0 ± 7.6 b | 19.7 ± 3.3 a | 29.8 ± 2.6 a | 136.4 ± 12.9 a | 11.8 ± 1.8 a |
|  | OWS | 1.7 ± 0.8 a | 11.6 ± 4.7 a | 56.7 ± 6.1 b | 15.9 ± 0.6 a | 31.4 ± 3.2 a | 142.9 ± 18.1 a | 8.9 ± 1.2 a |
|  | F value | 1.859 | 0.897 | 9.279 | 0.533 | 0.317 | 0.095 | 1.108 |
|  | *p* value | 0.211 | 0.441 | 0.007 | 0.604 | 0.736 | 0.910 | 0.371 |
| UNIJ | RF | 1.3 ± 0.5 a | 10.2 ± 1.3 a | 46.9 ± 5.5 a | 58.5 ± 9.4 b | 58.6 ± 4.4 a | 185.7 ± 19.6 b | 34.6 ± 6.9 b |
|  | DI | 1.2 ± 0.6 a | 9.8 ± 1.0 a | 43.5 ± 4.6 a | 42.8 ± 4.6 a | 43.9 ± 3.4 a | 144.1 ± 9.8 a | 34.8 ± 4.5 b |
|  | OWS | 0.8 ± 0.6 a | 8.1 ± 1.3 a | 39.0 ± 4.3 a | 39.9 ± 5.6 a | 40.4 ± 10.0 a | 157.4 ± 17.1 b | 29.4 ± 0.6 a |
|  | F value | 0.936 | 0.465 | 1.586 | 17.168 | 0.124 | 29.924 | 31.036 |
|  | *p* value | 0.427 | 0.642 | 0.257 | 0.001 | 0.885 | ≤0.001 | ≤0.001 |
| HAB | RF | 0.3 ± 0.0 b | tr | 3.9 ± 0.8 ab | 0.7 ± 0.1 a | 5.3 ± 2.4 a | 5.9 ± 0.9 a | 4.6 ± 1.1 b |
|  | DI | 0.1 ± 0.0 a | tr | 2.2 ± 0.6 a | 0.8 ± 0.1 a | 6.5 ± 2.9 a | 17.2 ± 3.2 b | 3.2 ± 0.7 ab |
|  | OWS | ND | tr | 5.4 ± 0.8 b | 0.6 ± 0.1 a | 6.6 ± 1.2 a | 14.9 ± 1.4 b | 2.5 ± 0.3 a |
|  | F value | 30.160 |  | 4.658 | 0.549 | 1.494 | 9.548 | 4.377 |
|  | *p* value | ≤0.001 |  | 0.041 | 0.596 | 0.275 | 0.006 | 0.047 |

The same letter shows no significant difference in carotenoid concentration between water supply treatments of cultivars according to Tukey's HSD post-hoc test; ND: Not detected; tr: Traces; free caps: Free capsanthin; free zeax: Free zeaxanthin; caps ME: Capsanthin mono-ester; zeax ME: Zeaxanthin mono-ester; β-carotene: Beta-carotene; caps DE: Capsanthin di-ester; zeax DE: Zeaxanthin di-ester; HET, Hetényi Parázs; UNIK, Unikal; UNIJ, Unijol; HAB, Habanero.

Under RF conditions, free zeax amounts were found to be significantly ($p$ = 0.043) higher when compared to those of DI and OWS. However, in the other cultivars (UNIK, UNIJ, and HAB), free zeax amounts did not change. Water supply treatments did not influence zeax ME and zeax DE concentration in HET and UNIK. In addition, in UNIJ, zeax ME and zeax DE concentration under RF conditions were found to be higher when compared to DI and OWS. Traces of Zeax ME were found in HAB even though amount was below detection limit; nonetheless, a significantly lower amount of zeax DE was detected

in OWS when compared to RF. Besides, concentrations in HAB were in lower amounts when compared to the other cultivars.

The amount of β-carotene in all cultivars was not affected by water supply treatments, and a very low concentration was recorded in HAB.

The sum of all individual carotenoid compounds that appeared on the chromatogram during the 2019 season showed that water supply treatments influenced some cultivars (Figure 8). Carotenoid concentration in HET under DI condition decreased significantly when compared to RF (F = 4.830, *p* = 0.038). There were no significant differences found among treatments for the other three varieties. Among cultivars, HET recorded the highest carotenoid concentration and lowest in HAB.

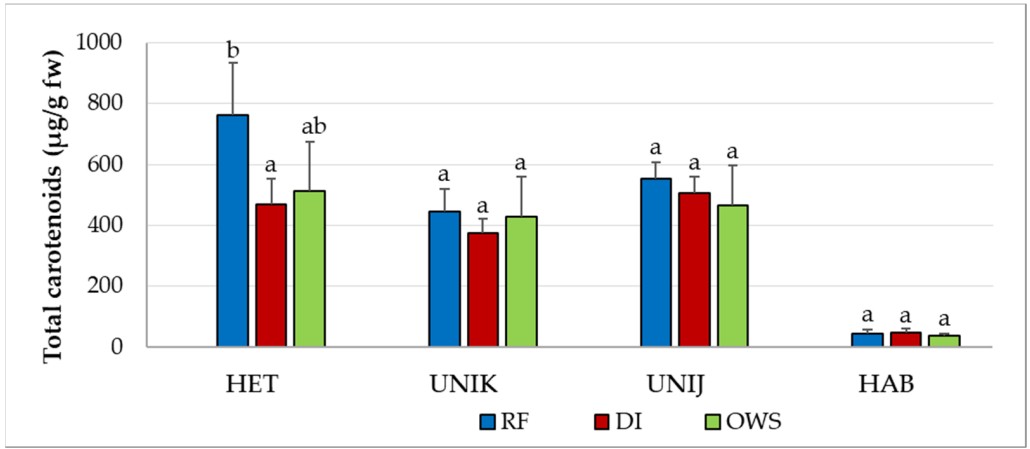

**Figure 8.** Effect of water supply on total carotenoid concentration in the 2019 season. The means are expressed in μg/g fresh base weight ± SD (n = 4). RF, rain-fed; DI, deficit irrigation; OWS, optimum water supply; HET, Hetényi Parázs; UNIK, Unikal; UNIJ, Unijol; HAB, Habanero.

## 4. Discussion

UNIK and HET cultivars had higher relative chlorophyll content during the 2018 season measurements under RF conditions when compared to DI and OWS and a further decrease in UNIJ and HAB (Figure 2A). We observed that HAB had a lower relative chlorophyll content during the pepper plants' growth period for the 2019 season (Figure 2B). UNIJ and HAB's genotypic structure as pungent peppers and their little or no photosynthetic pigmentation agree with previous studies as *C. annuum* cultivars showed similar behavior during their growth stages [30,31]. Cultivars that were rain-fed RF had higher chlorophyll content and a lower in the optimum water supply (OWS). This, however, explains further from our study that in 2019, less precipitation and irrigation favored SPAD values in cultivars, which measured the photosynthetic productivity and plant response to growth [32]. Due to high precipitation in 2018, the pepper plants had a lower relative chlorophyll content. Too much water supply caused a reduction in meristematic cell activity in the plant's growth, which eventually decreased the nitrogen uptake rate [33]. A decrease in cell expansion, which is also a contributing factor to relative chlorophyll content, was observed in previous studies of cotton plant leaves [34].

Leaf chlorophyll fluorescence expressed as the maximum quantum efficiency of PSII (Fv/Fm) was lower in HAB during the 2018 growth period under OWS. Water supply treatments did not influence Fv/Fm values in HET, UNIK, and UNIJ (Figure 3A). Our findings in the 2019 season also indicated a good response to Fv/Fm (Figure 3B). We measured a lower fluorescence ratio in plants after rainy days; however, there was no intraspecific variation in the pepper plants' fluorescence values. Optimal fluorescence ratio (Fv/Fm) values of leaves between 0.72–0.80 were found in this study. It confirms previous work on the ideal photochemical quantum efficiency of PS II when plants are not under any kind of stress [35]. Previous studies on 'Star flame' chili peppers are in agreement with this assertion [36]. A lower Fv/Fm values in HAB might be due to their

inability to withstand the open field environment. Demmig-Adams et al. [37] indicated that excess light absorption or light stress during plant leaf growth affects their response to high photon flux densities and PSII efficiency.

Canopy temperature reflects the plants' exposure to water stress. A lower canopy temperature in all cultivars was reported in OWS when compared to RF (Figure 4). The sensitivity of leaf stomata to water is necessary for leaf response to temperature [38]. We observed that as the water supply increased, the transpiration rate decreased in cultivars and a further decrease mostly in the 2018 season (Figure 4A), resulted in low canopy temperature and reduced growth. In addition, under DI conditions, a slightly lower canopy temperature was detected when compared to RF. Less irrigation and lower precipitation in the second growing season resulted in elevated temperatures in HET and HAB (Figure 4B). Besides, under elevated temperature as a result of elevated atmospheric carbon dioxide, these plants would have been expected to suffer stress, but that was not the case in this study [39]; nevertheless, UNIK and UNIJ with lower canopy temperature may result from leaf canopy's inability to absorb light due to water stress or excess water.

According to [40], vitamin C can be found in fully matured peppers and contribute essentially to human nutrition and health [41]. Based on the findings, water supply treatments significantly ($p < 0.05$) influenced vitamin C content in the chili pepper cultivars (Figure 5). Under RF conditions, a high amount of vitamin C was found in HET in 2018 (2204.2 $\pm$ 240.7 µg/g) and an increase in UNIK under DI conditions (2229.4 $\pm$ 146.3 µg/g). Similarly, an increase in vitamin C was found in UNIJ, but in HAB, water supply had no influence on their low amount (668 $\pm$ 163.7 µg/g).

The pepper cultivars had very high vitamin C content in the 2019 growing season as natural precipitation and irrigation given to plants was reduced. Generally, vitamin C content was lower in all cultivars under OWS (Figure 6). Higher content of Vitamin C was found in HET under RF condition (3371.5 $\pm$ 599.4 µg/g); however, the concentration decreased as irrigation increased (2326.9 $\pm$ 516.3 µg/g). HAB, on the other hand, recorded lower vitamin C amount in both RF (1659 $\pm$ 116.5 µg/g) and OWS (1344.7 $\pm$ 59 µg/g). Vitamin C oxidizes very fast when exposed to high to extreme temperatures [42]. Previous reports by Lee and Kader [43] support the findings of this study as low temperature and less precipitation in the 2019 growing season contributed to high amounts of vitamin C in the cultivars. Our results also showed a considerable effect of water supply treatments on DI peppers; a decrease in vitamin C was observed, which corresponds with a previous study by Ahmed et al. [44] when vitamin C content in 'Battle' hot pepper decreased at deficit irrigated conditions.

Regarding high amounts of vitamin C in cultivars under RF and DI but significantly decreased under OWS confirmed our findings that the use of little or no irrigation treatment in chili pepper cultivation could improve the sustainability of the water efficiency program [11]. Even though among the cultivars, HAB had the lowest vitamin C content, high amounts were also present in UNIK and UNIJ, which supports the assertion that vitamin C content in vegetables is mainly influenced by cultivars [40], rate of ripening [45], and seasonal conditions [46]. The low amount of vitamin C in HAB in 2018 and 2019 may be a result of their orange-red-like color attributes, which agrees with a previous study by Nagy et al. [29] when amounts of vitamin C was evident in red 'fire flame' hybrid peppers when compared to yellow colored.

The main composition of capsaicinoids responsible for pungency is capsaicin, and dihydrocapsaicin, as reported in the literature. These were found in higher amounts in all cultivars as well as nordihydrocapsaicin, which is usually characterized as a homolog [47], that were also present in higher concentration in the chili peppers (HET, UNIK, UNIJ, and HAB) during the 2018 season (Table 2). A higher concentration of CAP, which contributes to about 60% of pungency in peppers [17], was evident in HET (58.2 $\pm$ 14.4 µg/g), UNIJ (1282.7 $\pm$ 137.3 µg/g), and HAB (1822.1 $\pm$ 121.3 µg/g) under RF condition even though concentration did not vary among treatments. A higher pungency in RF peppers agrees with a previous study that less water results in capsaicin's stability in hot peppers [48] but

varies from cultivar [49], which further corresponds to the UNIK (17.3 ± 1.8 μg/g) cultivar under RF conditions in our study. Generally, lower levels of pungency were recorded in all cultivars when given OWS when compared to RF and DI. Low water supply and increased pungency in this research confirm a previous study by Jeeatid et al. [50] that less irrigation or water supply influences higher pungency levels in hot peppers. In UNIK peppers, CAP concentration decreased as water supply treatments increased.

A similar trend was observed in the 2019 growing season as the major capsaicinoids concentration were found to be higher in HET, UNIK, and HAB cultivars under RF conditions when compared to DI and OWS (Table 3). However, due to lower water supply in the second season (Figure 1), CAP levels in HET (405.1 ± 67.0 μg/g) and UNIJ (1662.5 ± 235.4 μg/g) under RF conditions and, in HAB (2969.7 ± 162.0 μg/g) under DI conditions were higher in concentration than in the first growing season. A general decrease in pungency as irrigation or water supply treatments increased was also evident in other pepper cultivars in this research. Studies have shown that capsaicinoid concentration changes under the various water supply treatments are usually attributed to uncontrolled environmental conditions [51].

The homolog compounds; HCAP, iDC, and HDCs, based on the results, were found in small quantities in both years and HCAP was absent in some cultivars due to changes in capsaicinoids accumulation behavior in peppers [52]. In this study, it was observed that lower capsaicinoid concentration in UNIK peppers, when compared to the other cultivars, might be due to their inability to withstand atmospheric light exposure or intensity, which contributes to a reduction in pungency in peppers [53]. Changes in pungency level in peppers in the 2018 and 2019 growing seasons based on this study, irrespective of water supply effect, varied among cultivars [54].

In this study, the peaks analyzed (free caps, free zeax, caps ME, zeax ME, β-carotene, caps DE, zeax DE) were characterized depending on their relevance in food chemistry. Capsanthin gives the primary red color of peppers, zeaxanthin represents the yellow color during ripening, and β-carotene is essential from a nutritional point of view [55].

Free caps concentration in HET decreased in OWS (26.5 ± 2.7 μg/g) when compared to RF or control (36.9 ± 5.4 μg/g) in 2018 (Table 4). During the second year (Table 5), lower precipitation and less irrigation resulted in low free caps concentration in HET under OWS (1.21 ± 0.4 μg/g) and DI (1.0 ± 0.3 μg/g) when compared to RF (1.9 ± 0.4 μg/g). Similarly, all other cultivars had low free caps concentration in the second year. The higher the level of free caps concentration in red peppers, the more they produce red color and aroma [21]. Generally, capsanthin amounts in the pepper cultivars, including the esters (caps ME and caps DE), decreased as water supply decreased even though the capsanthin esters present in some cultivars in this study were unstable when irrigated (DI and OWS). Free zeax concentration did not change in HET, UNIK, and UNIJ, and were found to be lower in HAB (0.4 ± 0.0 μg/g) under RF (Table 4). However, on the contrary to capsanthin concentration in the cultivar, which decreased dramatically in the second year (2019) under less precipitation, the concentration of free zeax increased in HET (15.1 ± 3.5 μg/g) cultivars instead and barely affected UNIK and UNIJ amounts (Table 5). Generally, water supply treatments did not significantly affect the monoesters of zeaxanthin (zeax ME and zeax DE) in some cultivars.

As referenced in the literature, β-carotene is a crucial component of carotenoids mostly found in vegetables such as red peppers [56]. Beta-carotene concentration did not vary in HET, UNIK, and HAB, even though a slight decrease was observed in these cultivars under OWS conditions. However, under DI (40.1 ± 1.6 μg/g) and OWS (37.1 ± 1.9 μg/g) conditions, concentration in UNIJ significantly decreased when compared to that of RF (64.1 ± 2.7 μg/g) (Table 4). A similar trend in β-carotene concentration was observed in the second growing season (Table 5). A decrease in β-carotene under DI and OWS conditions in cultivars may be a result of decreased precipitation and irrigation or water supply treatment, which may trigger the presence of P-cryptoxanthin, antheraxanthin, and violaxanthin, which contribute to the rapid synthesis of keto xanthophylls during

pepper fruit ripening [57,58]. As the water supply increased, β-carotene concentration decreased. Carotenoid concentration in HAB was very low throughout the research, and this is in agreement with a previous study that low concentration of carotenoids during fruit ripening remains low [59]. HAB had orange-red color attributes when compared to other cultivars. The color attributes of HAB confirm a previous study that indicated that total carotenoids are higher in red peppers than in yellow peppers [60].

Irrespective of the cultivar, total carotenoids were found to be higher under RF when compared to DI and OWS in both growing seasons. In view of the 2018 season, UNIJ cultivar accumulated higher carotenoids and lower in HAB (Figure 7). In addition, in the 2019 season, a slight decline in β-carotene was evident in cultivars as the water supply increased. Beta-carotene was higher in HET and lower in HAB (Figure 8). The degradation or changes of carotenoid concentration in the plant season as observed in this study may be a result of oxygenation, which is a major determinant in carotenoid degradation, particularly when pepper fruits become sensitive to light, heat, and oxygen [61].

## 5. Conclusions

Water supply to plants is a necessary component that contributes to crop growth. From this study, the use of less or no irrigation contributed greatly to the growth performance of the chili peppers, and this practice may benefit areas with water scarcity or shortage. Increased water supply or irrigation decreased relative chlorophyll content in the chili pepper plants. In addition, reduced precipitation and irrigation increased Fv/Fm in RF cultivated peppers. The growing of hot or spicy peppers under uncontrolled environmental conditions can affect their growth rate since unexpected rainfall or higher precipitation in the first season generally caused little physiological responses in the cultivars. It was evident that UNIJ and HAB performed poorly under open-field conditions, mostly in the 2018 season. However, cultivars' stable response in the second season proves that managing a good and smaller irrigation practice for pepper cultivation under an open field environment is achievable. Nevertheless, the selection of genotype for breeding should consider pepper crops that can withstand an environment with less water and not affect their vitamin C and carotenoid concentration as Habanero could not perform well.

Based on our findings, cultivars responded well under less water supply in the second season for vitamin C and capsaicinoids. The results also showed that individual carotenoids concentrations under increased water supply were higher in some cultivars. Due to pungency stability under little or no irrigation application except for precipitation, UNIJ hybrid pepper is recommended for consumer preference and pharmacological purposes. HAB, in this case, is recommended for pharmacological purposes only since water stress or optimum water supply may influence poor fruit setting and quality; however, consumers who are interested in their pungent attributes may consider it. HET, throughout the study, showed a very stable response and thrived well under open-field conditions. As such, it is highly recommended to breeders and growers and for consumer preference. UNIK is equally suitable for consumer preference under lower water supply management during their growth period.

To contribute to water management in crop cultivation and avoid water losses without any compromise on phytonutrients, future studies into these pepper cultivars will consider water supply treatment under modified atmosphere or greenhouses.

**Author Contributions:** Conceptualization, L.H., H.G.D., and S.A.D.; methodology, Z.P., A.N., and Z.N.; software, S.A.D.; validation, L.H. and Z.P.; formal analysis, S.A.D.; H.G.D., and Z.N.; investigation, S.A.D. and C.S.eS.; resources, L.H.; data curation, S.A.D., H.G.D., S.V., and Z.N.; data analysis, S.A.D. and S.V.; writing—original draft preparation, S.A.D.; writing—review and editing, L.H., H.G.D. and Z.P.; visualization, S.A.D.; supervision, L.H. and H.G.D.; funding acquisition, L.H. All authors have read and agreed to the published version of the manuscript.

**Funding:** The entire experiment was supported by the EFOP3.6.3-VEKOP-16-2017-00008 and EFOP-3.6.1-16-201600016 projects. This research was supported by the Ministry of Innovation and Technology within the framework of the Thematic Excellence Programme 2020, Institutional Excellence

Subprogramme (TKP2020-IKA-12) Stipendium Hungaricum/Tempus Public Foundation, and the Scholarships Secretariat, Government of Ghana.

**Institutional Review Board Statement:** Not applicable.

**Informed Consent Statement:** Not applicable.

**Data Availability Statement:** Data available on request due to Institutional restrictions and privacy.

**Acknowledgments:** Authors would like to thank everyone who contributed to this article, field, and laboratory work.

**Conflicts of Interest:** The authors declare no conflict of interest.

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
