# Peer review of "Effect of Water Supply on Physiological Response and Phytonutrient Composition of Chili Peppers"

_water, doi:10.3390/w13091284_

Round 1

Reviewer 1 Report

I found that the paper was substantially enhanced.

However, the form in which the paper in presented needs a strong revision.

Here some examples are given:

- In the abstract, the meaning of the cvs ID must be reported.

- In the materials and method section, the full name of the capsaicinoids evaluated must be reported.

- In tables and figure, the meaning of cvs and treatments ID must be reported.

- the format of figures 2, 3 and 4 must be standardize with that of figures 5 and 6. The same in true for figures 7 and 8.

- wrong use of capital letter for capsaicinoids’ ID.

- be careful, the term “between” refers to 2 things and “among” to more than 2 things.

- in the tables, there still is the same problem. While, in general, the letter “a” refers to the highest value, but still in some case “b” is attributed to the highest value (highlighted in blue in the PDF file).

- the result section must be improved. In particular, authors often describe in the details results that are not significant. The form also must be enhanced. The is no need to write “…however there were no significant differences between them” in all cases. That is implicit as only the significant differences should be reported. Also authors use to write “…concentration did not change” meaning that concentration are not different among treatment. This is incorrect as “change” should indicate variation over time.

- The paper needs a professional English editing.

Other note or corrections are reported in the PDF file.

Reviewer 2 Report

Effect of Water Supply on Phytonutrients of Chili Peppers – second revision

General:

The authors revised the manuscript according to several suggestions in my previous review. Pomological data and yields of various treatments and cultivars are still not reported but the reserach aim was phytochemical and therefore it seems that basic measurements have not been analyzed in the course of the trial. In the future, I suggest a more thorough research design.  The manuscript has been edited and rechecked for grammatical inconsistencies and the level of English has been improved. Only minor modifications are needed and in my opiniun the manuscript is now suitable for publication. 

Title: The authors considered my suggestion and changed the title to Effect of Water Supply on Physiological Response and Phytonutrient composition of Chili Peppers

Abstract:  The abstract is clear and sums up the most relevant results and adequately informs the reader on the methodology. Improvements to the text were made; however, the last sentence still needs of English editing. I suggest:

L28-30: Even though cultivars responded to the water supply treatments differently, HET exhibited a more uniform and stable composition in all treatments.

Keywords: the term ecological response was replaced by physiological response according to my suggestion

Introduction: The introduction has been edited and the text is now much more concise. The authors conformed to the suggested alterations for example in L32-33,  L36-37,  L42-44 etc. The paragraph in L 70-74 should be moved to the end of the Introduction and modified to:

Based on the results of the study, recommendations on suitable water supply for various pepper cultivars in light of their ability to withstand stress conditions without compromising their phytochemical composition have been transferred to the breeders, producers and food-industry in Hungary and other countries.

Materials and methods: Additional information on the management of the trial have been presented. The planting distance was reported, and the experimental design has been clarified. Differences in the amount of total water received between OWS (2019) and DI (2018) were explained. The number of plants of each repetition, the number of repetitions and the number of fruits included in biochemical protocols were added. Statistical analysis was rewritten and is much more precise.

Results and Discussion:

Several segments of the Results have been modified and are now improved, also because the authors clearly stated the effect of water treatments according to a single year. The level of English has been upgraded. In the Discussion section most dilemmas have been resolved, for example different respond of chlorophyll fluorescence in 2018 and 2019 and carotenoid composition. Another paragraph at the end of the Discussion adequately addressed these discrepancies. 

Round 2

Reviewer 1 Report

After the second review by the authors, some problems were resolved but other remains.

The problems remain and mostly in the Result section.

In situations where differences are not significant, authors still discuss data too much. However, I myself shortened several sentences.

In some cases, description of data was not in accordance with the values reported in the tables.

There still are mistakes in the attribution of letters to values (according to Tukey test).

Here they are:

Table 3.

CAP in UNIJ cultivar.

The ranking is: 1796 > 1662 > 1513 (DI>RF>OWS). It is not possible that the attributed letters are: b-a-b. It could be RF=1662a; DI=1796a; OWS=1513b.

Another option is that there is a mistake in the numbers!

Table 5.

I found errors in caps ME and UNIK. Here the problem seems that “a” is attributed to the lowest value instead to the highest and vice versa for “b”.

Caps ME in Het had the lowest value in DI (33.9). How is possible that the attributed letters are ab?

A possible and plausible option could be:  RF=57.5 a; DI=33.9b; OWS=37.7ab.

About caps ME in HAB I found both the previous problems. Inversion of letters and attribution of “ab” (which should indicate intermediate values), attributed to the lower value.

A possible option could be:  RF=3.9ab; DI=2.2b; OWS=5.4a.

Zeax DE in UNIJ.

Here it should be RF=34.6 a; DI=34.8a; OWS=29.4b.

Caps DE in HAB.

Here there might be a simple inversion of letters.

They should be: RF=5.9b; DI=17.2a; OWS=14.9b.

At last,  free zeax in HAB: why here “0.00” are indicated instead of “ND”?

Other issues are:

- in figures 2, 3 and 4 0%, 50% and 100% are indicated as ID for the treatment. This is confusing as in all other tables and figure “RF, DI and OWS” (and in the text) are reported. I suggest to go back to these treatments ID.

- in the caption of all tables and figures, Cultivar IDs and treatment IDs should be explicated (as it is in caption of figure 2).

Other comment/correction are reported in the attached PDF file.

Author Response

This manuscript is a resubmission of an earlier submission. The following is a list of the peer review reports and author responses from that submission.

Round 1

Reviewer 1 Report

The manuscript aims to evaluate the effect of water supply on ecophysiological response and phytonutritional composition of four chili pepper cultivars.

Although the results are not surprising, indeed they are quite obvious, in literature there is a lack of scientific evidence. Therefore, the topic is interesting and the manuscript deserves attention.

However, there are too many issues that make the manuscript not acceptable.

First of all, the Materials and Methods section is absolutely inadequate.

The treatments applied: they are defined as: rain-fed control, deficit irrigation (DI) and optimum water supply (OWS) well irrigation. However, what DI and OWS actually mean and how they were managed were never declared. Was, in OWS, restored the ETe? …and in DI restored the 50% of the ETe? Did that occur on weekly base?

In the introduction, the authors rightly said that the response of plants also depends on fertilization. However, how fertilization was handled was not fully reported. Was there a basal dressing? How much N, P2O5 and K2O were provided? The authors reposted that the top-dressing was performed via fertigation. Do this mean that the control did not have a top-dressing? And that the DI treated pepper had a half of the fertilizer of the OWS? If so, the author cannot say that the results depend on the water treatments but both, water and mineral nutrition. However, more information of fertilization must be reported.

The experiment layout is absolutely unclear. We know that the entire was one hectare wide, that design was a RCBD, but how many replicates (=blocks) were adopted; the size of the plots, and the planting distance (= actual plant density) were partially or totally missing, or they were unclear.

Data collection and analysis: for the ecophysiological determinations, how many leaves were taken per plant, and plants per replication (= block) were not always clear. For the chemical analysis, how many fruits were samples for each replication (is replication a synonym of block?) is not known. We just know that in the figures, “n= 4” is reported.

Statistical analysis: in lines 212-214 is written “The importance of variations between ecological responses on cultivars and water supply treatments was checked on a one-way basis using the Kruskal-Wallis test.” Then in lines 214-218 “The study of difference (ANOVA) between the effect of water supply treatments on ecological responses (..), vitamin C, capsaicinoids (…) and carotenoids (…) among cultivars (…) and water supply treatments (…) were followed by general linear model fitting test”. What does this mean? K-W test is a non-parametric method, does this mean that when data failed the homogeneity test the K-W test was performed and the ANOVA test when homogenous?

Looking at the figures I can think that the K-W test was performed on ecophysiological data and ANOVA + Tukey on phytonutrientional ones. Is that correct? And, were the ecophysiological data collected in different weeks pulled all together? Is it because of this that ecophysiological data failed homogeneity test? Everything is unclear in this section and remain unclear in the result section as well.

Table 6 should be discussed right after Table 5, likewise figure 7 should be discussed after figure 6.

Also is unclear why authors did not perform a 3-way factorial analysis as they have three factors:  year (variable factor), cultivar (variable factor), water irrigation treatment (fixed factor). This is very important.  . . also because, for instance, in the results section authors often report that results were different in the two years and there were differences among cultivar. Actually if you don’t perform a statistical test, you are not allowed to make those comparisons.

Result section. As previously said, sometimes differences between years and among cultivars are reported, but the statements are not supported by a statistical test as declared in the M&M section.

The format of Figures 3, 4 and 5 are inappropriate.  Line chart indicates a trend, and is indicated to describes variation over time, or in representations where increasing doses of a factor are used. Instead, there are no relationship between cultivars, so there is no reason to use lines to connect data. A column graph is more appropriate.

Throughout the paper, "low" and "high" instead "lower" and "higher". In some cases, the sentence might be grammatically correct, but this is improper in a scientific text.   

Little attention was paid to the representation of the results. All tables have errors. The letters representing Tukey's test results made me desperate. In some case “a” denote the highest value, in other the lowest. In some case it seems that “ab” was used instead of “a” or “b”. in still others, letters do not agree with the values (at a certain point I stopped to reed results because it has too hard to know what was right and what was wrong. Also, within each column, the same number of decimals should be used. In Furthermore, in the captions some information is missing.

At last, the conclusion section should be re-written because some sentences seem better discussions than conclusions, some others cannot be written just because there is no information on plant productivity to support the statements. Remember that the “conclusions” are simply the answer to the questions formulated in the objective of the manuscript. This was not the case in this manuscript. Actually, even if the objective of the research concerns the physiological and phytonutritional attributes of chilli pepper, a simple table on total chili pepper productivity at the beginning of the results section would be very important, as it would improve discussion.

For other minor comments, see the pdf file.

Reviewer 2 Report

Dear Authors

The manuscript topic meets the Journal aims. The relevant aspects of the topic are present , the use of other scientific literature could be considered to improve the paper quality.

Major revision should be taken, I believe that large parts of the manuscript should be rewritten. I strongly suggest the authors to take into consideration the following remarks.

About the quality of written English, the manuscript needs a language revision in order to improve its clarity and synthesis.

  • Abstract. the scientific scope of the study must be more highlighted. In addition, a concise summary of the key findings should be present.
  • Keywords have to be reconsidered, a couple of them are already present in the title
  • The introduction paragraph should be rewritten focusing on the subject, in my opinion underline on the water stress vs phytonutrients interaction. An Introduction should be clearer and more logical when it separates what the authors have done from what the paper itself covers.
  • Materials and Methods are in part adequately described, but in my opinion have to be check and, for some topics, rewrite in a more precise and clear way. All the agronomic part of the paper is almost missing
  • Results and discussion paragraphs follow the aim, experimental design and material and methods used. The results presented have to be revised and presented according to the improved material and methods and rewrite in clear form. Why the authors not represent and discuss the two years data together?
  • Conclusions needs to be much more succinct and have to be revised.

n. Line

note

22

“(μg/g)” not necessary

25

“chili peppers“ and “phytonutrients” words present in the title; HPLC is a method to me not specifically important as keyword

29

improve English readability

31

“flooding or waterlogging on the field” Soil flooding creates composite and complex stress in plants known as either submergence or waterlogging stress depending on the depth of the water table.

42

“for precise water, nutrient” maybe it could be changed to: ‘water and uniform application of water and nutrients direct..’

49

“Deficit irrigation” maybe a very short explanation could be useful; Deficit (regulated deficit) irrigation is one way of maximizing water use efficiency (WUE), the crop is exposed to a certain level of water stress either during a particular period (phenological stages) or throughout the whole growing season. We need to know the level of transpiration deficiency allowable without significant reduction in crop yield so the main objective of deficit irrigation is to increase the WUE of a crop by eliminating irrigations that have little impact on yield.

WUE is not present in your manuscript.

68

“Based on this study” what were the recommendations in the paper?

70

“water supply or irrigation” are synonyms, perhaps precipitation and irrigation

71-74

rewrite

78

“sandy loam, mostly cambisols with 1.6% organic matter and pH of 7.9” - Could you report more information on soil e.g. texture

82

“transported” transplanted

84

“Seedlings” at what phenological stage the seedlings were transplanted?

86

“three (3) for the three different water supply treatments rain-fed”;

for the three different water supply treatments to me there are not any clear description how they are applied, also regarding the use of the adopted FAO software.

Crop water requirement, irrigation requirement, irrigation water use and irrigation abstractions are often used synonymously without clear division.

To avoid confusion, we define these concepts briefly: crop water requirement (CWR) is the total amount of water required for transpiration by a well-managed crop grown under optimum growth conditions without water and nutrient stress.  For practical purposes, the CWR is calculated as the potential crop evapotranspiration (FAO  1996), avoiding the problem of clearly defining optimum growth conditions and optimum crop yield.

Irrigation water requirement is the amount of water that has to be applied in addition to rainfall to serve crop water requirements.

For irrigation planning it is determined as the difference between CWR (i.e. potential crop evapotranspiration) and the actual crop evapotranspiration under rain fed conditions with periods of water stress.

92

“was limited” what does it mean? it should be quantified

92

“Kc” insert ‘crop coefficient (Kc)’

97

“Irrigation was set-up using a drip system with weekly fertigation of nitrogen (NO3, NH4, NH2), phosphorus (P2O5), potassium (K2O), and sulphur (SO4) (Figure 2)” no information is present on the quality and quantity of the fertilizers used.

99

“Plant protection activities were carried out every week throughout the study.” no information is present

106

“based on fresh weight and good fruits for phytonutrient analyses” clearly express the criteria adopted for both weight and phytonutrient content

111

“Table 1.” unclear table, please note that the methodology applied for water returns has not been explained. In the first year the DI has values higher than the optimal one of the following year (2019); so define the deficit irrigation.

The distribution of precipitations in relation also to the phenological phase of the crop is not described.

To me the Meteorological data (tab and figures) should be discussed in the result paragraph.

114

“Figure 1.” for the two figures, with the exception of the axis of the ordinates of the temperatures, different amplitudes of the units relating to mm of rainfall and duration have been adopted, if possible to unify. Furthermore, it would have been interesting to be able to compare the trends in a single figure or in two figures by dividing the temperatures

119

“accordingly and adjusted” … it has not been, in my opinion, explained in the material and method, please provide

121

“Physiological responses” try to be more precise with the phenological phases. For example, a recent paper deals with the subject proposing a new description and codification of phenological growth stages of Capsicum spp. according to a modified BBCH scale (e.i. Feldmann, F., & Rutikanga, A. (2020). Phenological growth stages and BBCH-identification keys of Chilli (Capsicum annuum L., Capsicum chinense JACQ., Capsicum baccatum L.). Journal of Plant Diseases and Protection, 1-7.)

125-129

Simplify the description of the spad that is too articulated, the spad is a well-known device

146-150

“This portable battery-powered instrument is capable of …. while using the instrument.” Please Simplify the description

156-161

“About 3 grams of homogenized pepper fruit (seed excluded) was… into an HPLC column” if the protocol present in the bibliography has been adopted (Nagy et al.) it is absolutely not necessary to describe it accurately in the text.

227

“(RF or control, DI or deficit irrigation, and OWS or optimum water supply” all abbreviations and related explanations should have been made in the material and method so use only abbreviations

234

“low in OWS compared to RF” in a scientific text do not use low or high, rather quantify the differences and present them in a mathematically and statistically correct way as written in the following lines. Please rewrite.

270-272

“HAB had high canopy temperature; however, there was a difference in canopy temperature response between HET, UNIK and UNIJ.”

 The temperature of the canopy can be influenced by the width of the vegetative cover, are the data of the different genotypes land cover available?

Maybe measuring the leaf area index a dimensionless quantity that characterizes the canopies of the plants would have been appropriate

284

“vintage” crop year

303

“was found to be high under RF (control) conditions in HET and UNIJ, 303

and significantly (p<0.05) low in OWS.”  rewrite: was found, under RF conditions, be significantly (p<0.05) lower in OWS than in HET and UNIJ.

321

“table 3” why not represent and discuss the data in a single table and work out the two years together? as done for vitamin C.

329

“All major compounds were present in high concentration in all cultivars and minimal amounts in the homologues (Table 4)” discuss the data specifying the year.

562

“Conclusion” The Conclusion section presents the outcome of the work by interpreting the findings at a higher level of abstraction than the discussion and by relating these findings to the introduction stated. In this paragraph please do not simply summaries the points already made in the text instead, interpret your findings motivating the sentences.

Reviewer 3 Report

General:

The trial design is clear as is the aim of the study. The analytical part is adequate, but I feel that the study lacks some pomological and technological data. For example, the yield of the cultivars under each irrigation treatment should be recorded at harvest, the size of the fruit and the number of fruits on each plant should be presented. These put the biochemical results in perspective to plant performance under stress (drought) conditions. Moreover, physiological and biochemical studies should be interwoven with technological performance of plants as the aim of chili cultivation is not only ensuring optimal fruit composition, but also high yields of plants grown in open-field conditions. The manuscript should be edited and rechecked for grammatical inconsistencies and the level of English should be improved. I suggest some improvements in my review below.

Title: Informative but consider changing to Effect of Water Supply on Physiological Response and Phytonutrient composition of Chili Peppers

Abstract:  Clear and concise; it informs the reader on the trial setup, the cultivars, experimental conditions (treatments), and most relevant results. However, there are several grammatical inconsistencies which need to be resolved prior to potential publication. For example, I suggest the following improvement:

L13 Four pepper cultivars ('Hetényi Parázs', 'Unikal', 'Unijol' and 'Habanero') were subjected to different water supply treatments (RF rain-fed, DI deficit irrigation, and OWS optimum water supply) and cultivated in open field conditions.

Other minor changes would improve the level of English in this section.

Keywords: the term physiological response is more appropriate than ecological response

Introduction: The overall structure is good, perhaps some sentences are redundant as the authors tend to present the information in a somewhat lengthy manner. The introduction needs editing to make the text more concise. The aim of the study is clear.

Again, I must stress that the text needs English editing. Some suggestions to the text:

L32-33: Chemical and physical features of the soil (moisture and nutrient content) may also stress the plants and cause anatomical and physiological disorders.

L36-37: High temperatures and water deficit may cause oxidative stress and modify the synthesis of carotenoids in plants, growing in stressful environments.

L42-44: Drip irrigation is an effective technological measure in vegetable crop production as it enables precise water and nutrient application as well as reduce water loss due to evapotranspiration [7,8,9].

L44-45. The sentence is superfluous.

etc.

Materials and methods: Please specify the differences in the planting distance among cultivars used in the trial. The experimental design is not clear, particularly in terms of water treatments: how did the authors set the amount of water in deficit irrigation in relation to control? Ifthis treatment with low amounts of water. How did the authors set these dosages? If we compare the amount of total water received between OWS (2019) and DI (2018) the latter is higher! Please comment. How many plants constituted each repetition and how many repetitions were there in each treatment? How many fruits (n=?) were analyzed in biochemical protocols?

Phytochemical methods are adequate and detailed. Statistical analysis is suitable and it is only relevant to compare the treatments of a single year as the conditions were so dissimilar.

Results and Discussion: Some sentences are awkward and should be rewritten to improve these segments of the manuscript. For example, the sentence in L229-231 is unclear. Why do the authors state that the physiological measurements were recorded in 2018 in spite high precipitation? Were those not some of the basic parameters recorded in the study?

In 2018 it seems that higher water levels improved chlorophyll fluorescence and in 2019 reduced it (Figure 4). Why? The effect of water supply on canopy temperature (Figure 5) is an expected consequence of microclimatic alteration due to evaporation. It seems that each cultivar exhibited a different trend in vitamin C synthesis in relation to water supply in 2018. In the discussion the authors mostly stated that vit C content decreases at deficit irrigation conditions. But the results (particularly in 2019 and RF treatment) are contradictory to that.

The results on capsaicinoids clearly show up-regulation of these compounds in stress conditions. No differences in HDCs were detected among treatments in 2018 (a wet year) but significantly more HDCs were accumulated in fruit of RF and DI treatments. This seems plausible and the results are discussed with ample references on the topic.

Conclusion: The most relevant results are presented but it can be shortened and made more concise.